# A Sage–Husa Prediction Algorithm-Based Approach for Correcting the Hall Sensor Position in DC Brushless Motors

**DOI:** 10.3390/s23146604

**Published:** 2023-07-22

**Authors:** Lu Wang, Yong Cheng, Wei Yin

**Affiliations:** College of Energy and Power Engineering, Shandong University, Jinan 250061, China; wang_sgd2017@foxmail.com

**Keywords:** binary hall sensor, brushless DC motors, deviation correction, pre-calibration, Sage–Husa

## Abstract

Accurate knowledge of the rotor position is essential for the control of brushless DC motors (BLDCM). Any deviation in this identification can cause fluctuations in motor current and torque, increase noise, and lead to reduced motor efficiency. This paper focused on a BLDCM equipped with a three-phase binary Hall sensor. Based on the principle of minimum deviation, this paper estimated the relative installation offset between the Hall sensors. It also provided a clear method for ideal phase commutation position recognition and eliminated the Hall sensor installation position deviation. The proposed pre-calibration method identified and eliminated the offset of the permanent magnet poles, the delay time caused by the Hall signal conditioning circuit, and the offset of the sensor signal identification due to armature response under different loads. Based on the pre-calibration results, a correction strategy for correcting the rotor position information of BLDCMs was proposed. This paper presented a self-adaptive position information prediction algorithm based on the Sage–Husa method. This filters out rotor position information deviations that are not eliminated in pre-calibration. Experimental results on a hydrogen circulation pump motor showed that, after the pre-calibration method was adopted, the Mean Square Error (MSE) of motor speed fluctuations decreased by 92.0%, motor vibration was significantly reduced, average phase current decreased by 62.8%, and the efficiency of the hydrogen circulation pump system was significantly improved. Compared to the traditional KF prediction algorithm, the Sage–Husa adaptive position information prediction algorithm reduced the speed fluctuation during the uniform speed operation stage and speed adjustment stage, the speed curve overshoot, and the commutation time deviation throughout the process by 44.8%, 56.0%, 54.9%, and 14.7%, respectively. This indicates a higher disturbance rejection ability and a more accurate and stable prediction of the commutation moment.

## 1. Introduction

The potential of hydrogen fuel cells as an alternative energy source to internal combustion engines is increasingly being recognized, prompting a surge of research in this field [1]. Consequently, certain automakers have highlighted the hydrogen circulation pump, an integral component of the hydrogen fuel cell subsystem, as a significant area for in-depth study [2]. Owing to their superior power density, compact design, and impressive dynamic performance, brushless DC motors (BLDCM) are frequently selected as actuators within hydrogen circulation structures in a variety of vehicle control systems. When compared to other types of BLDCMs used in vehicular applications, those implemented within hydrogen fuel cell systems are marked by their high-speed operation and stable working attributes.

Inaccurate positional signals can lead to fluctuations in current torque, negatively impacting motor efficiency and generating mechanical vibrations and noise [3]. Therefore, obtaining accurate rotor position information is paramount for controlling BLDCMs. There are generally two methods for collecting this information through different algorithms: 1. Sensorless Algorithm: This includes techniques such as the back electromotive force method (BEFM), freewheeling diode method, inductance method, flux linkage observation method, and state observer method [4,5]. The essence of these methods is to detect physical quantities like voltage, current, and BEFM during motor operation and to estimate the rotor position information based on these motor parameters. However, these methods are influenced by factors such as changes in motor parameters, motor temperature, and any non-linear characteristics of the drive system [6], making it challenging to achieve control over the full range of motor speeds using only one algorithm [7]; 2. Position Sensor Method: This method involves using a position sensor to provide rotor position information. Common sensors include photoelectric encoders, rotary transformers, induction synchronizers, and linear or switchable Hall effect sensors. Although photoelectric encoders, rotary transformers, and induction synchronizers can directly provide rotor position information with high accuracy, they increase the cost and complexity of the system, reduce system stability and reliability, and have high environmental requirements [8,9]. Linear Hall effect sensors provide analog signals corresponding to the rotor position, but the signal contains harmonics, making it difficult to accurately determine the position information [10]. Binary Hall effect sensors can provide discrete position signals based on the edge features of the signal. With the appropriate estimation algorithm, accurate rotor position information that meets precision requirements can be obtained [11]. Therefore, some hydrogen circulation pumps are equipped with binary Hall effect sensors as the primary means of obtaining position information in their BLDCM systems.

Indeed, during long-term motor operation, the motor windings and other components may experience changes in their characteristics [12]. Temperature fluctuations and magnetic saturation are two factors that can cause motor parameters to deviate from their constant values [13]. These changes can impact the performance and efficiency of the motor. Additionally, the installation of position sensors may not be perfect, leading to unavoidable deviations between the actual and ideal sensor positions. These deviations can result in deviations that might affect the motor’s stable operation.

In a BLDCM, the accurate determination of rotor position information by binary Hall sensors can be affected by three primary types of factors: Type I: These factors include deviations and interference signals linked to the system itself, such as those arising from inaccuracies during the Hall sensor’s installation, deviations resulting from the uneven demagnetization of the permanent magnet, and those induced by changes in the motor’s parameters due to long-term operation. These deviations remain constant in size and direction, assuming identical testing conditions, and can be determined through theoretical analysis [14]; Type II: These factors involve separable interferences in the time domain that can be filtered out directly using time windows. While essential, these interferences are not the focus of this paper and thus will not be discussed in detail; Type III: These factors involve interferences that coexist with the sensor characteristic signals. They are difficult to discern in the time domain due to their random nature.

This paper’s main focus lies in discussing rotor position deviations and interference in the Hall sensor signal, considering the condition, “the 3-phase Hall sensors function normally”. The analysis did not consider situations where “certain phases or multi-phase hall sensors do not have signal output [15]”. Deviations and interferences under these conditions, hence, were not accounted for in this study.

As we dug deeper into the research by various experts in the field, it was clear that different strategies have been proposed to address deviations and errors introduced by the Hall sensor installation, demagnetization of permanent magnets, and motor parameters’ changes due to prolonged operation. Ritik Agarwal [16] developed a motor model using Simulink to determine if there exists an offset of the position sensor and quadrature error of the permanent magnet in a permanent magnet synchronous motor (PMSM) used in an elevator scheme with a 4-phase HALL sensor. The model explores the influence of sensor mounting deviation and the quadrature deviation of the permanent magnet on the motor control parameters. Eleven characteristic parameters were chosen as indicators to evaluate deviations. Mengji Zhao [17] discussed that, when controlling a PMSM using position information from 3-phase Hall sensors with positional deviation, the motor’s phase current waveform might showcase certain harmonic orders. To resolve this, a Luenberger observer was designed to rectify the sensor’s positional deviation. The effectiveness of this method was validated at a lower speed of 1200 rpm. Joon Sung Park [18], in his approach, calculated the angle difference between the edges of the 3-phase Hall sensor signals to gauge the relative positional deviation. Using the direct detection method of terminal voltage, the zero-crossing point of the BEMF was obtained. This point was delayed by 30 electrical degrees and compared with the Hall sensor signal’s edge, providing an estimate of the Hall sensor’s installation position deviation. The data were used to correct the 3-Hall sensor installation position. Dimitrios A [19] performed experimental research to investigate how an increasing relative positional deviation in the B-phase of the Hall sensor could impact motor bus current. This issue was addressed by utilizing the third-order harmonic characteristic of the bus current to portray the deviation amplitude and estimate the positional deviation. Hyun-Soo Seol [20] introduced an additional magnetic ring with a gap into a BLDCM, which was equipped with a three-phase Hall sensor, to simulate changes in the uniformity of the motor’s permanent magnets’ magnetic flux density. A two-dimensional finite element analysis was used to gauge the impact of the gap size on the permanent magnet rotor’s magnetic field distribution. Based on this, the Hall sensor deviation for different levels of magnetic flux density uniformity was determined and a corresponding deviation correction strategy was conceptualized. Liu Gang [21] tackled the problem of relative positional deviation between Hall sensors by performing a Fourier transform on the Hall sensor position signal. High-frequency interferences were filtered out, resulting in an improvement in the position signal’s estimation accuracy.

The referenced literature extensively investigated errors introduced by inherent system deviation, such as Hall sensor installation offsets, demagnetization of permanent magnets, and changes in motor parameters due to prolonged operation. Yet, several issues continue to prevail:Empirical theory suggests that the theoretical commutation point lies 30° past the zero crossing of the back electromotive force (BEMF). This same postulation can be utilized to estimate deviations in the installation position of Hall sensors. Nevertheless, real-world deployments reveal that fluctuations in rotational speed cause inaccuracies in the estimation of this 30° electrical angle, thereby undermining the precision of the inferred sensor installation position deviations;In the process of estimating deviations in the Hall sensor installation positions, signals obtained from actual measurements often necessitate filtering through Fourier series, making computational procedures appreciably sophisticated and difficult to directly implement during standard motor operation;In the context of a brushless DC motor in a hydrogen circulation pump typically operating at rated rotational speeds well exceeding 8000 rpm, it becomes essential to analyze certain deviations that could be overlooked at lower speeds.

Therefore, a comprehensive analysis of the deviations in the installation position of the three-phase Hall sensors, pole displacement, the delay inaccuracies induced by the Hall signal conditioning circuit, and the Hall sensor signal displacement spurred by armature reaction is imperative. Further, the substitution of the conventional terminal voltage method with the Line-BEMF method will facilitate a superior approximation of sensor installation position deviations.

Random disturbances intertwined with sensor signals are challenging to separate directly in the time domain, which affects the acquisition of rotor position information [22]. Therefore, it is necessary to employ effective measures to suppress noise in the rotor position information and obtain accurate position data. Corresponding algorithms primarily include the sliding mode observer, extended Kalman filter, Luenberger observer, artificial neural networks, and fuzzy logic [23,24].

Peter Billeschou [25] meticulously engineered a torque observer predicated on Luenberger’s principles, simulating the generation of phase current noise by instigating load oscillations on the output shaft of a brushless direct current motor (BLDCM). The empirical evidence underscores the observer’s stability in estimation and its swift responsiveness to torque fluctuations. Jose-Carlos Gamazo-Real [26] employed a pair of tri-layered neural networks to estimate the rotor position and velocity of a BLDCM, wherein the phase resistance and inductance remained elusive. The efficacy and alacrity of the neural network methodology were juxtaposed with those of the sliding mode observer and the extended Kalman filter. The findings illuminated the neural network method’s superiority in terms of efficacy and speed. Xinyue Li [27] proposed a Kalman-based motor parameter identification methodology that takes into account the influence of the voltage source inverter (VSI) on the performance degradation of the VSI and the alterations in motor parameters due to the protracted operation of the BLDCM system. The experimental results reveal that, compared with the Kalman motor parameter identification method that disregards the VSI influence and the traditional motor parameter identification method, the Kalman-based method is more precise. Aishwarya Apte [28] designed a two-tiered observer comprising a sliding mode observer (SMO) and a disturbance observer (DO) to estimate the speed and torque of a motor, respectively. The reference value for the observer system equation’s parameter was procured through table lookup predicated on the motor’s real-time operating conditions. This value was utilized to enhance the convergence speed of the cascaded observer. Experimental results indicate that the proposed observer controller significantly ameliorates the motor’s dynamic performance compared to the proportional-integral (PI) controller. Furthermore, owing to the characteristics of the SMO and DO, the algorithm operates with remarkable speed and simplicity.

In practical applications, motor parameters evolve over time, necessitating corresponding adjustments to certain system model parameters within the observer. Some researchers opt for an adaptive filtering method based on the Kalman filter [29] to suppress the noise in the rotor position signal. The adaptive filter, while filtering measured data, concurrently estimates some system model parameters, utilizing limited, indirect, and noisy measurements to infer information that is challenging to measure or ascertain directly [30]. The adaptive filtering method further encompasses the output error method, innovation method, Sage–Husa method [31], strong tracking filtering method, and adaptive robust and second-order [32] mutual difference method [31], among others. These methods have been effectively employed in the engineering field. Provided that all parameters of each predictive algorithm are optimized, the choice of any one would suffice. However, in the face of uncertain parameters and persistent random errors that are difficult to filter out, the Sage–Husa method emerges as a commendable choice. It estimates positional information while simultaneously estimating system noise expectation q, system noise variance Q, measurement noise expectation r, and measurement noise variance R, thereby accelerating the estimation speed while minimizing estimation fluctuations [33].

Given the characteristic of the Sage–Husa method, where the accuracy of historical information directly influences the precision of the prediction, it is crucial to minimize any bias in the historical data to ensure more accurate prediction results. Therefore, one approach could be to first use pre-calibration to correct the sensor position information, and then employ the predictive algorithm to obtain accurate motor commutation position information. This process would help to ensure that the Sage–Husa method has the most accurate data from which to make its predictions, thereby improving the overall accuracy of the system.

The implementation of pre-calibrated corrections substantially mitigates bus current and rotational speed fluctuations in electric motors, which arise from various phase shift deviations, consequently enhancing motor efficiency. By incorporating the Sage–Husa method into the motor control system, an innovative algorithm for predicting phase shift information, capable of tracking multi-system noise parameters, has been proposed. Compared to traditional forecasting techniques, this approach exhibits superior resistance to disturbances, ensuring more precise and stable predictions for the moment of phase transition.

This paper ventured into an in-depth investigation of the brushless DC motor utilized in a hydrogen circulation pump, equipped with a quintessential binary Hall sensor. The remainder of this paper is organized as follows: The notorious issues of sensor installation deviation, characteristic of three-phased Hall sensors, magnetic pole displacement, delay aberrations owing to the Hall signal conditioning circuitry, and the Hall sensor signal deflections consequent to the armature reaction were all embraced within the ambit of this study. Intrinsic examinations of these deviations and their rectification methodologies were performed. The Line-BEMF method served as a conduit for the identification and calibration of whimsical sensor installation deviations. Concurrently, pre-calibration protocols were established for the other residual deviations (Section 2). Subsequently, the system model for the brushless DC motor utilized in the hydrogen circulation pump was constructed. The avant-garde Sage–Husa algorithm was incorporated into the phase-prediction facet of the DC motor (Section 3). Our terminus lies at the exposition of the pre-calibration experiment and the revelation of the comparative experimental outcomes standing at the juncture of the Sage–Husa paradigm and the conventional forecasting algorithms (Section 4).

## 2. Pre-Calibration of Rotor Position Signal Deviation

In a BLDCM system equipped with a three-phase binary Hall sensor, the most fundamental control approach entails utilizing all Hall sensor edges as motor commutation indicators to govern motor operation. These discrete signals encompass a myriad of system deviation s and disturbances, chiefly comprising Hall sensor installation position deviations, magnetic pole offsets, armature reaction-induced Hall sensor signal shifts, and signal conditioning circuit delays. Through pre-calibration, pertinent offsets are duly rectified.

Upon the uniform operation of the BLDCM, if the three-phase Hall sensors are installed with an electrical angular disparity of 120 degrees, the phase variance of the respective Hall signals ideally also amounts to 120 degrees, with a duty ratio for Hall sensor signals at 50%. Consequently, the curve for positional estimation obtained therefrom is a linear function of constant gradient. Figure 1 schematically illustrates the phase relationship of the Hall sensor signal edges with deviations and ideal conditions, as well as the angular position estimation results.

In Figure 1, $X_{r}$ and $X_{f}$ denote the ascendant and descendent demarcations of the Hall sensor signal exhibiting deviation, where ‘X’ could signify A, B, or C, representing the pertinent parameters of the three phases of the BLDCM. $X_{r}^{’}$ and $X_{f}^{’}$ symbolize the ascendant and descendent edges of the archetypal Hall sensor signals. The electrical angles of the skewed ascendant and descendent demarcations are designated as $\theta_{X-r}$ and $\theta_{X-f}$, respectively. Conversely, the ideal electrical angles of these edges are denoted as $\theta_{X-r}^{’}$ and $\theta_{X-f}^{’}$. The electrical angular disparity between the biased edges and the ideal edges is represented as $\varphi_{X-r}=\theta_{X-r}-\theta_{X-r}^{’}$ and $\varphi_{X-f}=\theta_{X-f}-\theta_{X-f}^{’}$. The projected rotor position signal is computed by preserving solely the 0th and 1st-order of Taylor series expansions, whilst presupposing a current rotational velocity congruent to the preceding stage’s mean velocity.

### 2.1. Relative Position Deviation between Hall Sensors

Presuming an absence of eccentricity discrepancies in the three-phase Hall sensor’s implementation, solely non-uniform distribution remains. To assess the sensor’s installation deviation, we posit a sextet of equidistant demarcations throughout an electrical angle cycle. These conjectural boundaries are ascertained by minimizing the cumulative squared phase disparity between each sensor’s periphery. Disregarding extraneous factors and assuming $\varphi_{X}=\varphi_{X-r}=\varphi_{X-f}$, the interspacing amid the ascending borders of Hall sensor phases difference between A-phase with B-phase and B-phase with C-phase may be deduced via the subsequent calculation:(1)$$\left\{\begin{array}{l} \theta_{B-r}^{’}-\theta_{A-r}^{’}=(\theta_{B-r}^{}-\theta_{A-r}^{})+\varphi_{A}-\varphi_{B}=2\pi /3\\ \theta_{A-r}^{’}-\theta_{C-r}^{’}=(\theta_{A-r}^{}-\theta_{C-r}^{})+\varphi_{C}-\varphi_{A}=2\pi /3 \end{array}\right.$$

The aggregate squared deviation can be denoted as $S=\varphi_{A}^{2}+\varphi_{B}^{2}+\varphi_{C}^{2}$. Upon attaining the minimum value for this summation, we observe $\delta S/\delta \varphi_{A}=0$. Incorporating this outcome into Equation (1) and postulating that the phase deviation between respective edges are $\varphi_{\min X}$, a positive deviation signifies a measurement signal edge trailing the ideal boundary. Consequently, the following relationship emerges:(2)$$\left\{\begin{array}{l} \varphi_{\min A}=(2\theta_{A-r}-\theta_{B-r}-\theta_{C-r})/3\\ \varphi_{\min B}=(2\theta_{B-r}-\theta_{A-r}-\theta_{C-r}-2\pi )/3\\ \varphi_{\min C}=(2\theta_{C-r}-\theta_{A-r}-\theta_{B-r}+2\pi )/3 \end{array}\right.$$

From this analysis emerges an optimal location for the hypothetical dividing line, evaluated further by the relative positioning deviations $\varphi_{\min X}$ along each sensor edge. At this juncture, should an absolute positional deviation be discerned for any phase Hall sensor signal edge, it can conceivably provide an estimate for the installation deviations of each sensor edge.

### 2.2. Absolute Position Deviation of the Hall Sensor

The juncture of commutation within the two-phase conduction control paradigm of the brushless direct current motor (BLDCM) epitomizes the optimal boundary of the Hall sensor signal. Assuming a negligible impact from cogging, magnetic saturation, eddy currents, and hysteresis losses, and in the absence of armature reaction, it becomes feasible to evaluate the commutation point through a sensorless position-estimating algorithm, thereby obtaining an impartial measurement. This estimation, when contrasted with the Hall signal, facilitates the discernment of the positional discrepancy in the installation of each sensor edge. To achieve this, we have chosen to utilize the Line-back electromotive force (Line-BEMF) algorithm. Comprehensive elucidation of the algorithm is available in reference [34]:(3)$$\left[\begin{array}{l} e_{AB}\\ e_{BC}\\ e_{CA} \end{array}\right]=\left[\begin{array}{l} e_{A}-e_{B}\\ e_{B}-e_{C}\\ e_{C}-e_{A} \end{array}\right]=\left[\begin{array}{l} u_{A}-u_{B}\\ u_{B}-u_{C}\\ u_{C}-u_{A} \end{array}\right]-\left[\begin{array}{ccc} R_{A} & -R_{B} & 0\\ 0 & R_{B} & -R_{C}\\ -R_{A} & 0 & R_{C} \end{array}\right]\left[\begin{array}{l} i_{A}\\ i_{B}\\ i_{C} \end{array}\right]-\left[\begin{array}{ccc} L_{MA} & -L_{MB} & 0\\ 0 & L_{MB} & -L_{MC}\\ -L_{MA} & 0 & L_{MC} \end{array}\right]p\left[\begin{array}{l} i_{A}\\ i_{B}\\ i_{C} \end{array}\right]$$
where $u_{X}$ represents the voltage of each individual phase with respect to the ground, $e_{X}$ denotes the BEMF voltage of phase *X*, and $e_{XY}$ signifies the Line-BEMF voltage between phases *X* and *Y* (where X,Y∈{A,B,C} and X≠Y), $L_{MX}$ corresponds to the equivalent inductance of the stator coils for each phase. This specific value is derived from the disparity between the self-inductance and mutual inductance of the phase in question. Additionally, $i_{X}$ serves as the phase current, $R_{X}$ is the resistance attributed to each phase, and p embodies the differential operator.

Presuming that the ascending edge of Hall A-phase functions as the reference boundary and that the proximate zero-crossing point of the BEMF aligns with the zero-crossing juncture of the descending $e_{BC}$ segment, the phase discrepancy between these occurrences epitomizes the absolute position deviation of Hall A-phase’s rising edge. Throughout the conversion from A-phase and C-phase conduction to the *AB* two-phase conduction, *A* represents the non-switching phase, *B* signifies the phase to be conducted, and *C* denotes the phase to be terminated. As the switching point nears, the current remains unaltered, implying that $p(i_{C})\approx 0$. The induced current in phase *B* approximates zero, namely $i_{B}\approx 0$, thereby simplifying the equation as follows:(4)$$e_{BC}=u_{B}-u_{C}-R_{B}i_{B}$$

The phase angle corresponding to the zero-crossing juncture of $e_{BC}$‘s falling segment is designated as $\theta_{LEMF}$. The installation position deviation of Hall sensor A-phase rising edges comprises two components: the first entails the relative positional deviation, while the second involves the dividing line’s deviation, ascertainable through the phase difference between the corresponding dividing line of the Hall sensor A-phase rising edge and $\theta_{LEMF}$. Let $\varphi_{PosX-r}$ and $\varphi_{PosX-f}$ represent the installation deviations of the rising and falling edges, respectively. The deviation constitutes the disparity between the sensor edges’ angle and the ideal commutation angle, calculable via Equation (5).
(5)$$\left\{\begin{array}{l} \varphi_{PosA-r}=\theta_{A-r}-\theta_{LEMF}\\ \varphi_{PosB-r}=\varphi_{\min B}+\varphi_{PosA-r}-\varphi_{\min A}\\ \varphi_{PosC-r}=\varphi_{\min C}+\varphi_{PosA-r}-\varphi_{\min A} \end{array}\right.$$

Upon solely accounting for the installation deviation of the Hall sensor, three correction values for the installation deviation associated with the falling edges, denoted as $\varphi_{PosX-f}$, can be procured through the relationship $\varphi_{PosX-r}=\varphi_{PosX-f}$.

### 2.3. Deviation of Permanent Magnet Poles

Hall sensor signals are intrinsically linked to the motor rotor’s magnetic field. Inconsistencies in the permanent magnet’s remanence, installation angle deviation of the magnetic pole pieces, or rotor eccentricity can result in deviations of the permanent magnet poles. For a BLDCM operating at a steady speed, a phase difference exists between the three-phase Hall sensor signals under steady-state operation, though the signal characteristics remain consistent. Considering a 2-pole BLDCM, the ideal phase difference between each pole is 90° in mechanical angle. The magnetic pole deviation and motor magnetic pole schematic can be deduced by utilizing any one of the Hall sensor signals, as illustrated in Figure 2.

Envisioning a 360-degree mechanical angle as one complete cycle, two rising edges and two falling edges transpire within a single cycle, and the angular disparity between adjacent edges is π/2 in the absence of magnetic pole deviation. Employing Hall signal A-phase as an exemplar, with the rising edge corresponding to magnetic pole N1 as the cycle’s inception and the motor rotating in a clockwise direction, the edges are denoted as $\theta_{N1-r}$, $\theta_{N1-f}$, $\theta_{N2-r}$, and $\theta_{N2-f}$, respectively. Assuming no relative deviation exists between the N1 magnetic pole and the ideal position, the phase differences between the other three edges and the ideal edges are $\varphi_{N1-f}$, $\varphi_{N2-r}$, and $\varphi_{N2-r}$, respectively:(6)$$\left\{\begin{array}{l} \varphi_{N1-f}=\pi /2+\theta_{N1-r}-\theta_{N1-f}\\ \varphi_{N2-r}=\pi +\theta_{N1-r}-\theta_{N2-r}\\ \varphi_{N2-f}=3\pi /2+\theta_{N1-r}-\theta_{N2-f} \end{array}\right.$$

The absolute deviation of reference magnetic pole N1 has been pre-calibrated as mentioned earlier, allowing for the calculation of the remaining magnetic pole offsets. Taking into account the inconsistencies in the residual magnetism of permanent magnets, the angular deviation of the motor’s magnetic pole installation, and the rotor eccentricity, which primarily affects the motor’s slot torque with minimal impact on load torque characteristics, the presence of fan load can further mitigate the influence of slot torque on motor performance. Consequently, in practical applications, the evaluation of speed fluctuations induced by magnetic pole offsets should be considered. If the effect is negligible or torque ripples introduced by the offsets can be overlooked, the impact of this deviation may be disregarded.

### 2.4. Sensor Deviation Introduced by Armature Response

The magnetic field in BLDCM is subject to a myriad of factors, encompassing the performance of the permanent magnetic material, the magnetization technique of the magnetic pole, the dimensions and form of the pole shoe, the air gap’s length, and the armature’s axial extension. By amalgamating the armature reaction magnetic field with the primary magnetic pole field, the air gap magnetic field is derived. The progressive augmentation of the motor armature magnetic field in response to escalating loads necessitates careful consideration. Concurrently, an indeterminate displacement arises between the sensor signal edges and the theoretical commutation position. The erratic distribution of the actual rotor magnetic field further complicates the direct calculation of the Hall sensor’s offset under the influence of the armature reaction via theoretical analysis. In this context, calibration serves to acquire the correction value of the offset under varying current conditions.

Owing to the presence of specific rotor position discrepancies in the position data extracted from the sensorless approach, in light of the armature reaction, a magneto-electric position sensor was elected for rotor position identification. Under constant speed conditions, diverse rotor currents were employed to ascertain the rising and falling edge offset angles of the Hall sensor signal across an array of operational circumstances. A regression analysis facilitated the derivation of the relationship curve between the average phase current, denoted as $\bar{I}$, and the offset angle, delineated by $\theta_{OFS-rise}(\bar{I})$ and $\theta_{OFS-fall}(\bar{I})$.

### 2.5. Hall Signal Conditioning Circuit Delay

Output signals from Hall sensors necessitate filtration and conversion to facilitate interpretation by microprocessors. Depicted in Figure 3a, a typical signal conditioning circuit incorporates signal pull-up, passive RC filtering, and signal voltage regulation.

In Figure 3a, R1 signifies the sensor’s output impedance, R2 denotes the pull-up resistor, R3 corresponds to the limiting resistor, R4 represents the voltage divider resistor and the input impedance of the microprocessor capture port, and C1 is the filter capacitor. As the Hall sensor signal transitions from high to low, the frequency response of the RC circuit is characterized by $\tau_{C1}=(R1+R3)*C1$. In contrast, when the Hall sensor signal shifts from low to high, the associated frequency edge is marked by $\tau_{C2}=R2*C1$, exhibiting a higher value than $\tau_{C1}$. The Hall signal is approximated as a sequence of square wave signals replete with abundant high-order harmonics. Owing to the presence of filtering capacitors, the various harmonics display corresponding delays, which escalate as the rotational speed increases. The $\theta_{CKT-Rise}(n)$ and $\theta_{CKT-Fall}(n)$ curves delineate the relationship between the delay angle and rotational speed post-calibration. Figure 3b portrays the Hall signals before and after conditioning during motor operation.

### 2.6. Integration of Various Position Information Deviation Correction Strategies

In practice, each sensor signal discrepancy is individually calibrated during the offline phase or throughout motor initialization, subsequently being supplied to the controller for rectification during motor operation. The calibration procedure is illustrated in Figure 4.

Calibration of magnetic pole offset: this discrepancy can be executed at varying velocities, and the acquired measurements may be subjected to arithmetic mean computation. As the speed diminishes, the influence exerted by the load decreases. Consequently, the measurement outcomes can be weighted and averaged based on the motor’s velocity;Calibration of relative position discrepancy between the three-phase Hall sensors: this discrepancy can be calibrated during each initialization. Either rising or falling edges can be chosen and the corrective value can be ascertained based on Equation (2);Calibration of absolute position discrepancy: this discrepancy can be identified by selecting an edge and the nearest Line-BEMF zero-crossing point. Utilizing Equation (4), the requisite terminal voltage and real-time phase current signals can be measured. According to Equation (5), the absolute position discrepancy correction value for each edge can be computed;Calibration of signal conditioning circuit delay: this discrepancy necessitates a comparison between each Hall signal prior to and following conditioning, which can be calibrated during an offline phase without load. At disparate speeds, the edge delays of the signal conditioning circuit can be statistically procured based on the microprocessor input capture threshold;Calibration of Hall sensor signal offset induced by armature reaction: this discrepancy can solely be calibrated during an offline phase with a typical load. By orchestrating motor operation experiments under varying loads and amalgamating them with the previously acquired signal conditioning circuit delay, this offset can be ascertained. Given that the load torque of the hydrogen circulation pump is directly proportional to the square of the motor load and velocity, while the current corresponds directly to the motor output torque. Consequently, the relationship between the rotational speed and the discrepancy introduced by the armature reaction can be utilized in lieu of the association between the average phase current and the discrepancy induced by the armature reaction.

## 3. Prediction Algorithm of Rotor Position Information Based on Sage–Husa Method

During the motor’s operation, interference signals are intermingled with the Hall sensor signals, exhibiting a certain degree of randomness and posing difficulty in direct separation within the time domain. Hence, drawing upon the Sage–Husa approach, a motor commutation position prediction algorithm was devised to anticipate and track the commutation location, subsequently executing commutation in accordance with the predicted outcomes. For the hydrogen circulation pump-driven motor, the load can be approximated as a fan, with the resistance torque being proportional to the square of the electrical angular velocity.
(7)$$\left\{\begin{array}{l} \theta =\theta_{0}+\omega_{0}t+\varepsilon t^{2}/2\\ \omega =\omega_{0}+\varepsilon t \end{array}\right.$$
where ω denotes the electrical angular velocity, ε signifies the rotor angular acceleration, θ represents the rotor electrical angle, and the subscript 0 indicates the initial value of the corresponding variable. The mechanical equation of motion for the motor is provided as follows:(8)$$J\varepsilon =J\frac{d\omega }{dt}=T_{e}-T_{L}-f\omega$$
where J denotes the rotational inertia, $T_{e}$ represents the electromagnetic torque, which can be expressed as the product of torque coefficient $K_{T}(n)$ and average current I. $K_{T}(n)$ can be regarded as a coefficient related to velocity, thereby facilitating equation correction under specific operating conditions. Furthermore, $T_{L}$ represents the load torque, which can be described as $M(n)\cdot \omega^{2}$, where M(n) signifies the propeller characteristic coefficient associated with motor speed n. Given that motor speed does not experience abrupt changes, the average angular velocity $\bar{\omega }$ over a certain period can replace ω. f is the friction coefficient of the motion system; however, as the friction resistance of the motor is significantly smaller than the fan resistance within the hydrogen pump’s normal working speed range, it can be disregarded. If one commutation represents the commencement of a cycle, the motor’s motion state during a cycle can be denoted by X(k+1). Subsequently, the systematic equation $X(k)={\left[\omega_{k},\theta_{k},I_{k}\right]}^{T}$ can be derived from Equations (7) and (8):(9)$$\left[\begin{array}{c} \omega_{k+1}\\ \theta_{k+1}\\ I_{k+1} \end{array}\right]=\left[\begin{array}{ccc} 1 & 0 & \frac{K_{T}(n)}{J}\Delta T\\ \Delta T & 1 & \frac{K_{T}(n)}{2J}\Delta T^{2}\\ 0 & 0 & 1 \end{array}\right]\left[\begin{array}{c} \omega_{k}\\ \theta_{k}\\ I_{k} \end{array}\right]-\left[\begin{array}{c} \Delta T\\ \frac{\Delta T^{2}}{2}\\ 0 \end{array}\right]\frac{M(n){\left(\bar{\omega }\right)}^{2}}{J}+W_{k}$$

Given that ΔT represents the sampling interval, $I_{k}$ denotes the mean phase current value within the most recent sampling interval, and $W_{k}$ signifies the system noise matrix, we postulate that the system noise exhibits the properties of colored noise. The motor system is furnished with a trio of Hall sensors and three-phase current sensors, strategically positioned at 120° intervals, thereby facilitating the formulation of an observational equation:(10)$$Z_{k}=\left[\begin{array}{c} \omega_{k}\\ I_{k} \end{array}\right]=\left[\begin{array}{c} \begin{array}{ccc} 1 & 0 & 0 \end{array}\\ \begin{array}{ccc} 0 & 0 & 1 \end{array} \end{array}\right]\left[\begin{array}{c} \omega_{k}\\ \theta_{k}\\ I_{k} \end{array}\right]+V_{k}$$
where $Z_{k}$ signifies the observation at cycle k. $V_{k}$ represents the measurement noise matrix, it is further assumed that the measurement noise exhibits the characteristics of colored noise.

Assuming that the colored noise $W_{k}$ and $V_{k}$ can be expressed as the sum of white noise and an offset, the system noise $W_{k}$ has an offset $q_{k}$ and variance $Q_{k}$, while the observation noise $V_{k}$ has an offset $r_{k}$ and variance $R_{k}$, with both noises being uncorrelated. The statistical characteristics of the noise can be represented as $E[W_{k}W_{j}^{T}]=Q_{k}\delta_{kj}$, $E[V_{k}V_{j}^{T}]=R_{k}\delta_{kj}$, $E[W_{k}]=q_{k}$, $E[V_{k}]=r_{k}$, and $E[W_{k}V_{j}^{T}]=0$, where $\delta_{kj}$ denotes the impulse, and. Referring to the relevant equations presented in literature [35], the filtering equations can then be formulated as follows:(11)$$\left\{\begin{array}{l} {\hat{X}}_{k,k-1}=\Phi_{k,k-1}{\hat{X}}_{k-1}+{\hat{q}}_{k-1}\\ P_{k,k-1}=\Phi_{k,k-1}P_{k-1}\Phi_{k,k-1}^{T}+{\hat{Q}}_{k-1}\\ {\tilde{Z}}_{k}=Z_{k}-H_{k}{\hat{X}}_{k,k-1}-{\hat{r}}_{k}\\ {\hat{X}}_{k}={\hat{X}}_{k,k-1}+P_{k,k-1}H_{k}^{T}{\left[H_{k}P_{k,k-1}H_{k}^{T}+{\hat{R}}_{k}\right]}^{-1}{\tilde{Z}}_{k}\\ P_{k}=[I-K_{k}H_{k}]P_{k,k-1} \end{array}\right.$$

In these equations, any variable A with an added superscript, $\hat{A}$, denotes the estimated value of that variable; $K_{k}$ represents the Kalman gain coefficient, which serves as a transitional variable; $P_{k,k-1}$ is the posterior error covariance at the optimal estimation during cycle k; $P_{k}$ refers to the prior error covariance during cycle k at the optimal estimation; and ${\tilde{Z}}_{k}$ signifies the residual (residue). To simplify the equations, let $X(k)=X_{k}$, $X(k+1,k)=X_{k+1,k}$, $Z(k+1,k)=Z_{k+1,k}$,$P(k|k-1)=P_{k,k-1}$, and $P(k|k)=P_{k}$.

Under the fulfillment of the following two conditions:The system matrix is fully controllable and fully observable, ultimately leading the system to a stable state, satisfying $\underset{k\to \infty }{\lim}P_{k}\to P$.The white noise components of the observations at each moment are uncorrelated, which is satisfied when $C(i)=E[(Z_{k,k-1}-r_{k})(Z_{k-i,k-i-1}^{T}-r_{k-i})]=0,(0<i<k)$, the equations are given as:(12)$$\left\{\begin{array}{l} {\hat{r}}_{k+1}=(1-d_{k}){\hat{r}}_{k}+d_{k}(Z_{k+1}-H_{k+1,k}{\hat{X}}_{k+1,k})\\ {\hat{R}}_{k+1}=(1-d_{k}){\hat{R}}_{k}+d_{k}({\tilde{Z}}_{k+1}{\tilde{Z}}_{k+1}^{T}-H_{k+1,k}P_{k+1,k}H_{k+1}^{T})\\ {\hat{q}}_{k+1}=(1-d_{k}){\hat{q}}_{k}+d_{k}({\hat{X}}_{k+1}-\Phi_{k+1,k}{\hat{X}}_{k})\\ {\hat{Q}}_{k+1}=(1-d_{k}){\hat{Q}}_{k}+d_{k}(K_{k+1}{\tilde{Z}}_{k+1}{\tilde{Z}}_{k+1}^{T}K_{k+1}^{T}+P_{k+1}-\Phi_{k+1,k}P_{k}\Phi_{k+1,k}^{T})\\ d_{k}=(1-b)/(1-b^{k+1}),0<b<1 \end{array}\right.$$

Theoretically, the Sage–Husa method is capable of estimating the expectation of system noise q, the variance of system noise Q, the expectation of measurement noise r, and the variance of measurement noise R. However, in practical applications, accurately tracking all four parameters concurrently proves to be arduous, potentially leading to imprecise or even divergent outcomes [36]. Furthermore, discerning the source of the deviation between the predicted state variable and the actual observed variable value, whether it arises from system or measurement noise, poses a significant challenge. Concurrent adjustment of multiple parameters may also diminish the algorithm’s robustness [37]. According to findings presented in reference [38], the error spectrum resulting from adaptively adjusting Q alone exhibits a flatter profile compared to simultaneous adjustment of both Q and R.

Considering that system noise typically varies with changes in application scenarios and environmental temperature, accurately obtaining Q can be difficult. Owing to these random factors, it is reasonable to assume that the initial value of the system error offset $q_{0}$ is approximately zero. Measurement noise, on the other hand, is generally a statistical parameter that can be calculated based on the corresponding measurement errors of the observed system signals. Hence, the present study intended to conduct theoretical analyses of the sensor variance for measurement noise variance R and measurement noise offset r, determining reference values for the initial noise parameters. Subsequently, an adaptive algorithm was employed to track Q. Once the value of Q stabilized after multiple tracking attempts, R, r, and q were tracked sequentially.

Assuming that the noise affecting rotational speed and current exhibits an uncorrelated, biased Gaussian distribution, let $R_{A}$ represent the variance of the rotational speed observation ω, and $R_{B}$ denote the variance of the average current observation I. By utilizing the average angular velocity ω obtained within the sampling interval, the primary sources of error, such as installation inaccuracies and conditioning circuit delays, were accounted for and have been addressed through pre-calibration. Consequently, the initial value of $R_{A}$ could be set to zero. Current values were obtained through current sensors, and the average value within a single phase-switching interval was considered. Digital filtering was applied to the data before averaging. The initial value of the measurement noise variance for the predictive algorithm could be set to $R_{0}=\mathrm{diag}(0,R_{I})$, where $R_{I}$ represents the error of the current sensor. Additionally, since the measurement noise being considered was random noise, the initial value of r could be set to zero.

## 4. Experiment and Discussion

### 4.1. Introduction of the Experimental Bench and Motor Control System

Experimental confirmation was undertaken on a specialized centrifugal hydrogen circulation pump testing apparatus. This apparatus primarily consisted of a BLDCM system, a controller, various sensors, and a power supply system, as well as a signal observation and data collection system. The BLDCM system operated at a rated voltage of 80 V, with an expected rotational speed of 15,000 RPM. It comprised a motor with two poles and an estimated phase resistance of 65.5 ohms, complemented by a three-phase Hall position sensor. The control system was bifurcated into two integral sections—the control board and the drive board. The control circuit featured an MC9S12XEP100 as its main chipset, and the drive circuit utilized an IR2108. The driver switch tube employed was an IPB200N25N3. To circumvent cumulative thermal damage to the controller by the full-bridge circuit, a corresponding heat sink was creatively designed. The system was equipped with a magnetic electrical sensor (SZMB-10) for precise measurement of the motor rotor’s position, along with a current sensor (LA25-P) dedicated to the measurement of phase current and bus current. The power supply framework primarily comprised a VARIEO-RU60-10060 for powering the driver and an RPB3003D-3 providing power to the current sensor. The signal observation and acquisition equipment comprised a Tektronix2014B oscilloscope and a ZTIC-EM9118 acquisition, as shown in Figure 5 and Table 1.

During the regular operation of the hydrogen circulation pump system, the target rotational speed remained relatively stable. Given that the fan load torque and speed have a quadratic relationship, the motor speed was controlled in a closed-loop manner by adjusting the drive system current using a PID algorithm. The BLDCM employs square wave driving, utilizing position signal correction strategies and commutation moment prediction algorithms to acquire the ideal location for each Hall edge, which serves as the sign for motor commutation. Experiments were conducted in the range of 5k rpm to 10k rpm to corroborate the feasibility of the algorithms proposed in this study.

### 4.2. Identification and Calibration of Partial Deviation

In a 360° mechanical angle, there are four edges for each of the three-phase Hall sensor signals. Assuming that two rising edges are $HallAR_{1}$ and $HallAR_{2}$, and two falling edges are $HallAF_{1}$ and $HallAF_{2}$ respectively, *X* can be *A*, *B*, or *C*.

Permanent magnet pole shift identification

According to Figure 4 (Correction Strategy), the identification of magnetic pole displacement was conducted, with the selection of signal from phase *A*. Considering it as the reference, the three remembered rotational speeds during the experiment, along with their corresponding statistical deviation values, were obtained and are presented in Table 2. The phase difference percentages between the remaining three edges and the ideal edge are also provided. In Table 2, the weights for the respective rotational speeds are assigned as 0.5, 0.3, and 0.2, respectively.

Due to the fact that the ideal rising and falling edges of the Hall sensor can be mutually deduced, it was possible to select any set of rising and falling edges to infer the rotor position information. According to Table 2, the phase deviation of $HallAF_{1}$ is larger compared to $HallAR_{1}$ and $HallAF_{2}$. To mitigate the speed fluctuations caused by this edge deviation, only the rising edge could be selected for deviation correction, and the falling edge can be deduced based on the correction results of the rising edge. The experimental results demonstrate a significant reduction in speed fluctuations using this approach.

2.Relative position deviation between Hall sensors

In accordance with Figure 4 (calibration strategy), the relative positional deviation between the three-phase Hall sensors was identified. The edge deviations were set as $\varphi_{\min X1}$ and $\varphi_{\min X2}$. The test results for the three speeds were also weighted and processed, as shown in Table 3. The deviation correction values in the table were used for calibrating the relative deviations between the sensors during normal motor operation.

The differences of various edges at different speeds in Table 3 are relatively small. Based on the relevant theory outlined in Section 2.1, the weighted average value was deemed appropriate as the correction value for the relative positional deviations among the sensors in the pre-calibration process.

3.Absolute position deviation correction

The reference signal chosen for measurement was HallAR. The voltage at the *B* and *C* phases, as well as the real-time phase current at the B phase, were measured. Taking the calibration stage at 1107 rpm as an example, Figure 6a displays the real-time voltage signals at the *B* and *C* phases, as well as the real-time phase current signal at the B phase. Figure 6b illustrates the relationship between the zero-crossing points of the line counter $E_{BC}$ and the Hall-A phase signal.

In accordance with the calibration strategy depicted in Figure 4, it is evident that there exists a certain hysteresis during the motor’s phase shift. The specific hysteresis angle can be calculated by determining the phase difference between the falling segment’s line-BEMF zero-crossing point and the Hall-A phase edge. In order to highlight the phase of the line-BEMF zero-crossing point more prominently, the calculated data of $1/E_{BC}$ has been included in the figure. The test results at the three speeds were also subjected to weighting and the outcomes are presented in Table 4. These correction values serve to rectify any sensor installation deviations during regular motor operation.

Similarly, in Table 4, the differences of various edges at different speeds are also minimal. Thus, following the framework presented in Section 2.2, the weighted average value was considered suitable as the correction value for the absolute positional deviations in the pre-calibration process.

4.Hall Signal Conditioning Circuit Delay

In accordance with the calibration strategy depicted in Figure 4, the results of the delay time adjustment circuit are illustrated in Figure 7a. The data in the figure represent the average of multiple measurement results.

From the graphical representation, it is evident that the observed pattern of delay correction aligns with the theoretical framework presented in Section 2.5. When applying this correction in practical applications, it is advised to obtain the required adjustment values through interpolation based on the corresponding speeds.

5.Sensor signal offset introduced by armature reaction

By installing a (30-1) encoder on the input shaft of the motor and using the magnetic electric sensor to provide reference position signals, the phase variation of the rising edge on Hall-A phase was calibrated under different loads. A calibration was performed at 800 rpm, obtaining mechanical angles of 311.7° and 131.8° for $HallAR_{1}$ and $HallAR_{2}$, respectively, which were taken as the reference values for zero armature reaction deviation. Testing was conducted within a speed range of 5000 rpm to 10,000 rpm. The correction angles introduced by the armature reaction are depicted in Figure 7b, where the data represent the average of multiple measurement results.

The graphical representation illustrates that the observed pattern of offset correction aligns with the theoretical framework presented in Section 2.4. When applying this correction in practical applications, it is recommended to obtain the necessary adjustment values through interpolation based on the corresponding speeds.

### 4.3. Online Calibration Experimental Stage

To validate the efficacy of each deviation correction technique, a comparative experimental investigation was conducted, employing the following methodologies: (I) A square wave was utilized to drive the BLDCM, while the full bridge circuit was propelled by a two-to-two conduction control approach. This method implemented 12 discrete steps per 360° mechanical angle and the speed range was precisely regulated using PID control. The duty cycle of the full bridge circuit switch tube served as the controlled variable, with the commencement of commutation marked by the Hall sensor edge; (II) The sole alteration in this setup pertained to the commutation point, wherein the corrected Hall sensor edge replaced the original point of commutation. Notably, the correction outcomes remained consistent across various speeds. Figure 8 graphically delineates the speed comparison during 10 mechanical cycles of the rotor, specifically measured at 5700 rpm.

The analysis shows that, after deviation correction, the Mean Square Error (MSE) of the speed is reduced by 8.0% compared to before and the speed fluctuation is significantly reduced, indicating a smoother operation of the motor.

Figure 9 illustrates the comparative outcomes of the terminal voltage signal and phase current signal. As depicted in the figure, the terminal voltage waveform prior to calibration exhibits evident characteristics of delayed commutation. Conversely, the waveform of the corrected terminal voltage and phase current closely approximates the waveform of the signal during commutation at the correct timing. Furthermore, at identical speeds, the peak current observed prior to calibration approaches 10 A. However, following the calibration, it diminishes to less than 4 A. Remarkably, the average current post-calibration stands at a mere 37.2% of its former value, resulting in substantial reductions in power consumption.

The comparative experimental results show that the average phase current after deviation correction is smaller, indicating higher motor efficiency, thus proving the effectiveness of the correction method.

### 4.4. Prediction Algorithm of the Phase Commutation Time Based on Sage-Husa

When predicting the phase commutation time based on (9)–(12), it is necessary to determine relevant parameters in the system model, such as the rotary inertia J, torque coefficient $K_{T}(n)$, and propeller characteristic coefficient M(n). Using experimental data from a fan-less loaded motor accelerating within the range of 800–3000 rpm, the rotary inertia of the motor system without fan-loaded can be derived by analyzing changes in voltage, current, and speed. Combined with the known design parameters of the additional gear encoder, the rotational inertia of the motor system can be calculated as $J\approx 0.0039\;\mathrm{kg}\cdot m^{2}$. In addition, a regression can be performed on the values of $K_{T}(n)$ and M(n) based on experimental data obtained from a fan-loaded motor running within the range of 800–10,000 rpm, where changes in voltage, current, speed, and magnetic sensor position signal were recorded.

In the observation matrix, let the initial value of parameter X be $X_{0}$, where $q_{0}=0$ and $r_{0}=0$ are chosen. Due to the precision of the current sensor, $R_{0}=\mathrm{diag}[0,0.02]$ is set. As the hydrogen circulation pump motor operates within the speed range of 5000–10,000 rpm, the prediction based on Sage–Husa begins with a stable speed of 5000 rpm. With the tracking process, $P_{k}$ quickly stabilizes. Since the covariance matrix $P_{k}$ is symmetric, a relatively large initial value can be chosen, and $P_{0}=\mathrm{diag}[50000,50000,50000]$ was selected here.

As the minimum adjustment step of the PWM module of the XEP100 chip is 1/240, and the power supply settings remain unchanged, at most 15-speed control points can be obtained within the range of 5000–10,000 rpm. For the purpose of presentation, seven points were selected, and the mean square deviation of the speed was taken as the evaluation parameter to assess the effectiveness of the deviation correction and prediction algorithms. The speed gradually increased, starting from 5000 rpm in the experiment and stabilized for 1 s near each test point. Among the four noise parameters, except for $Q_{k}$, the other three parameters can be roughly determined based on theoretical analysis. Therefore, the prediction algorithm mainly tracks $Q_{k}$, while $R_{k}$, $r_{k}$, and $q_{k}$ are selected for tracking at intervals of 10 turns (120 commutation cycles).

Figure 10 compares the predicted speed results of pre-calibration correction prediction (with a correction delay of 60° electrical angle as the next commutation position), the prediction based on the Kalman Filter (KF) algorithm, and the prediction based on the Sage–Husa method for the motor speed range between 5000 rpm and 10,000 rpm. The stability of the predicted speed indirectly reflects the accuracy of the commutation information prediction results.

By utilizing the speed values obtained from Figure 10, the commutation moments at each commutation position were inferred. A selected local time window was chosen and the commutation times for each method within this time period are displayed in Figure 11. For comparison purposes, the ideal commutation time data, obtained by smoothing the measured speed data, has been included.

The data from Figure 10 and Figure 11 were analyzed to assess the level of speed fluctuations and the accuracy of time predictions for the three prediction methods. The Mean Absolute Percentage Error (MAPE) was utilized to quantify the degree of error between the instantaneous speed and commutation time predictions and the actual values. The evaluation metrics chosen included the MAPE for the speed in the speed regulation stage, the maximum overshoot in the speed regulation stage, the MAPE for the speed in the constant speed stage, and the MAPE of the commutation time throughout the entire process. The results of this evaluation are compiled in Table 5.

Based on Figure 10 and Figure 11 and Table 5, it can be observed that, throughout the entire process, both the KF predicted speed curve and the Sage–Husa predicted speed curve exhibit smaller fluctuations compared to the speed curve solely based on pre-calibration correction. This suggests that both prediction algorithms are effective in tracking the motor speed. Although the majority of the Sage–Husa prediction results are similar to the KF prediction results, there are instances where the KF predictions exhibit larger fluctuations, resulting in discrepancies in speed and commutation time fluctuations compared to the measured and ideal commutation times.

From the analysis of the speed fluctuations, speed overshoot, and commutation time results in both the constant speed and speed regulation stages, it can be observed that the Sage–Husa results yield a reduction of 44.8%, 56.0%, 54.9%, and 14.7% compared to the KF predicted results. This indicates that the Sage–Husa algorithm achieves higher prediction accuracy, is less affected by random disturbances, and is closer to the ideal commutation results. Therefore, the position information prediction algorithm based on Sage–Husa is effective in utilizing historical information to predict ideal commutation times.

## 5. Conclusions

The BLDCM with binary Hall sensors relies on the edge detection of Hall sensor signals for commutation. However, if there is a deviation in the Hall sensor signals, it can lead to current and torque fluctuations, reducing motor efficiency and causing mechanical vibrations. In this paper, the deviation of Hall signals was classified into three categories: (1) delays and sensor recognition deviations caused by Hall signal conditioning circuits and armature reaction at different loads; (2) sensor installation and magnetic pole displacement; and (3) interference introduced into the Hall sensor signals during operation. Each type of deviation was analyzed and a correction method was proposed to address these deviations.

The main findings of the paper are as follows:Given that some of the deviations capable of causing displacement in motor position information necessitate consideration under high-speed operating conditions, yet can be disregarded under low-speed operating conditions, delays introduced by conditioning circuit latency, offsets introduced by armature reactions, installation deviations of Hall sensors, and pole offsets were each addressed by employing offline and online calibration methods for identification and subsequent utilization in correcting motor commutation positions. Empirical results illustrate that, in comparison to direct commutation using Hall sensor edges, employing the edges post commutation location correction for commutation reduces the phase current of the motor to 37.2% of its pre-correction state, and the rotational speed Mean Squared Error (MES) is diminished to 8.0% of its pre-correction value;The absolute installation position deviation of the Hall sensor was estimated utilizing the Line-BEMF, which, in comparison to the BEMF estimation, enhances the precision of position information estimation while simultaneously reducing the number of calculations;Capitalizing on the stable load characteristics of the hydrogen circulation pump, the Sage–Husa method was introduced into the motor control system, formulating a commutation moment prediction algorithm that can track multiple system noise parameters. This algorithm was employed to estimate the commutation position during operation. Compared with the traditional Kalman Filter (KF) prediction algorithm, the Sage–Husa adaptive position information prediction algorithm reduces speed fluctuations, overshoot of the speed curve, and commutation time deviation throughout the process by 44.8%, 56.0%, 54.9%, and 14.7%, respectively, during both the uniform operation stage and the speed adjustment stage. Consequently, it exhibits superior disturbance rejection capacity, offering more accurate and stable predictions for the commutation moment.

The design and realization of phase prediction algorithms for a hydrogen circulation pump utilizing the BLDCM based on the Sage–Husa method involve extensive theoretical, methodological, and technical realms, presenting several challenges that necessitate future endeavors to untangle them. Due to the operational characteristics of the hydrogen circulation pump, which involve a lower frequency of speed adjustment and stable load, there has been no application experiment of this algorithm during frequent speed adjustments. In the future, we will seek application scenarios with different characteristics to verify this algorithm and similar ones.

## Figures and Tables

**Figure 1 sensors-23-06604-f001:**
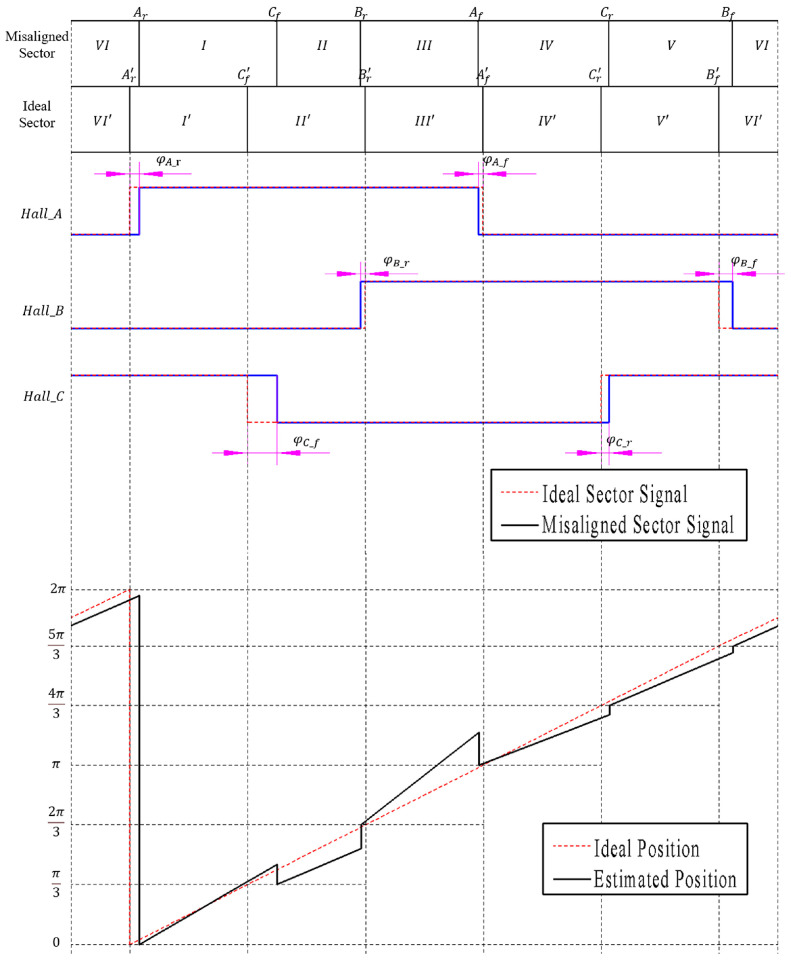
Schematic diagram of phase relationship of the Hall sensor signal edges with deviations and ideal conditions, as well as the angular position estimation results.

**Figure 2 sensors-23-06604-f002:**
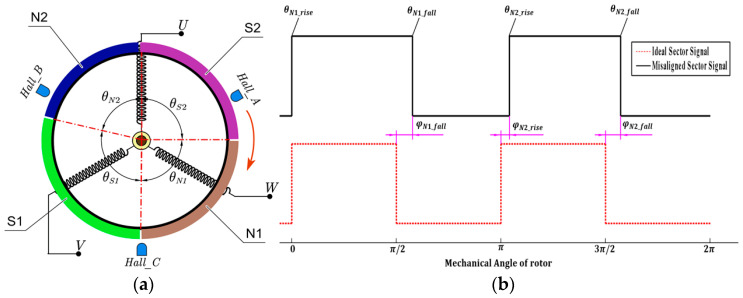
Schematic of the motor with a pole deviation (**a**) and the corresponding Hall signal (**b**).

**Figure 3 sensors-23-06604-f003:**
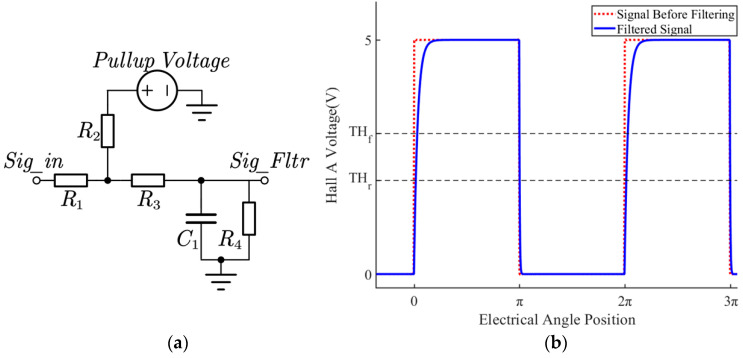
Schematic diagram of the Hall signal conditioning circuit (**a**) and the signal before and after conditioning circuit (**b**).

**Figure 4 sensors-23-06604-f004:**
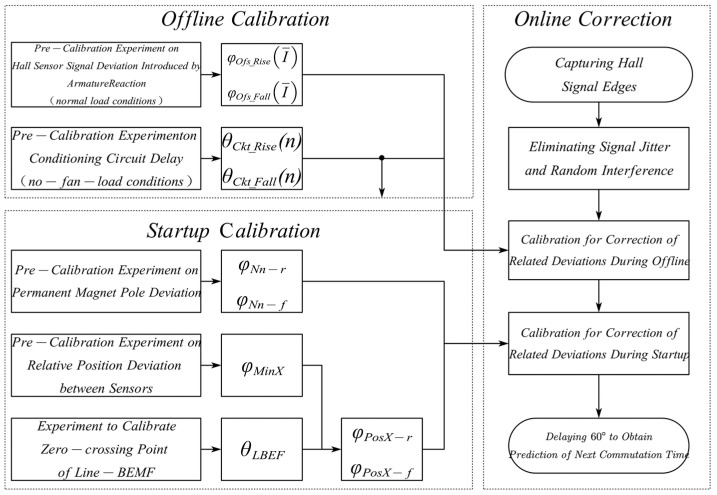
The flowchart for the integration of Hall sensor signal deviation correction strategies.

**Figure 5 sensors-23-06604-f005:**
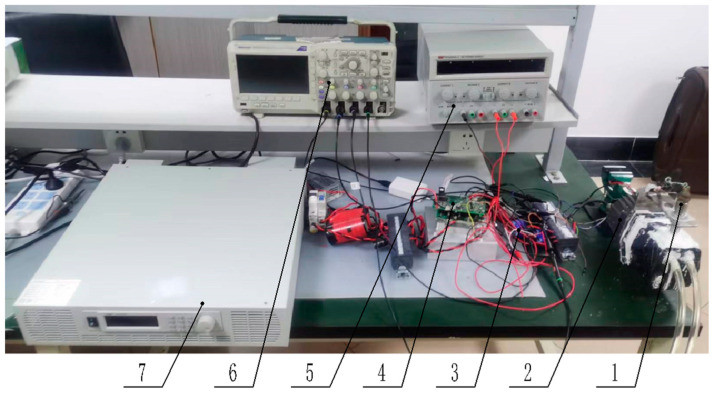
Some instruments and equipment of the experimental bench (1—Magnetoelectric sensor 2—BLDCM 3—Current sensor 4—Controller 5—RPB3003D-3 6—Tektronix2014B 7—VARIEO-RU60-10060).

**Figure 6 sensors-23-06604-f006:**
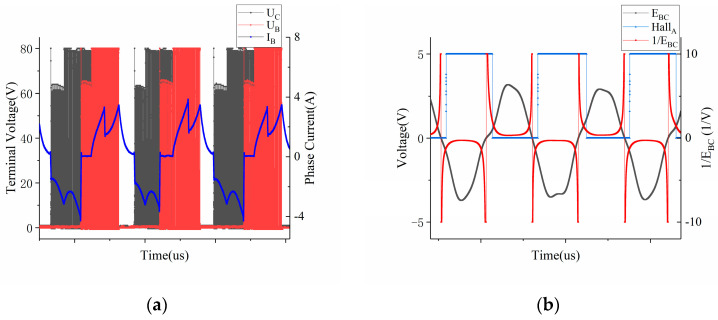
Terminal voltage and phase current related to *E_BC_* (**a**) Phase relationship between Line-BEMF and Hall−A phase (**b**).

**Figure 7 sensors-23-06604-f007:**
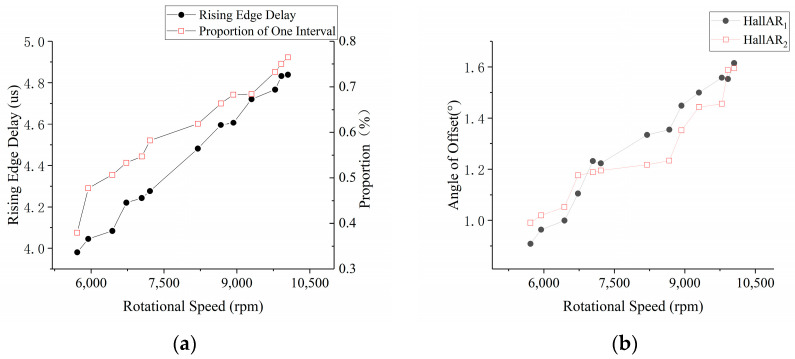
Delay angle of the conditioning circuit (**a**) and Offset angle introduced by armature response (**b**) at each speed.

**Figure 8 sensors-23-06604-f008:**
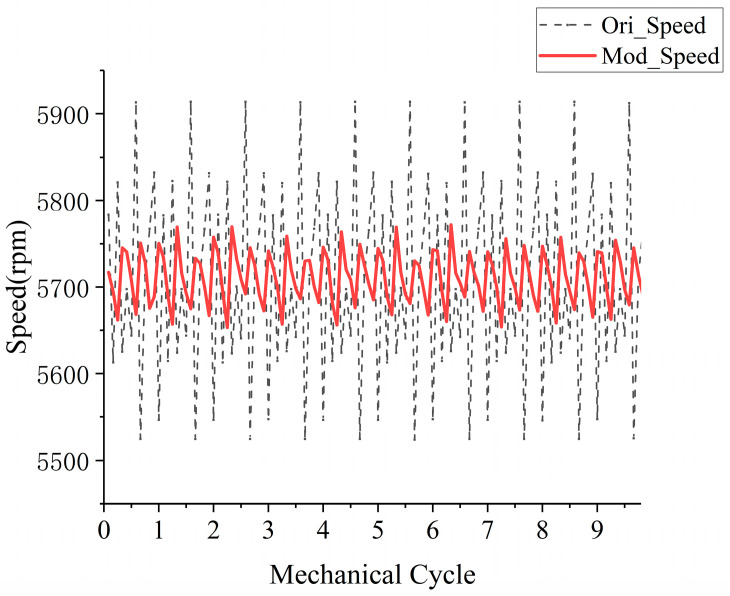
Comparison of speed fluctuation of original strategy and speed fluctuation of correction strategy.

**Figure 9 sensors-23-06604-f009:**
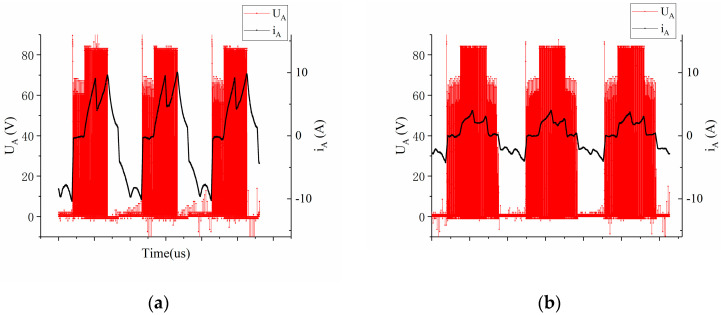
Before correction (**a**) After correction (**b**). Comparison of the terminal voltage signal and the real−time phase current signal.

**Figure 10 sensors-23-06604-f010:**
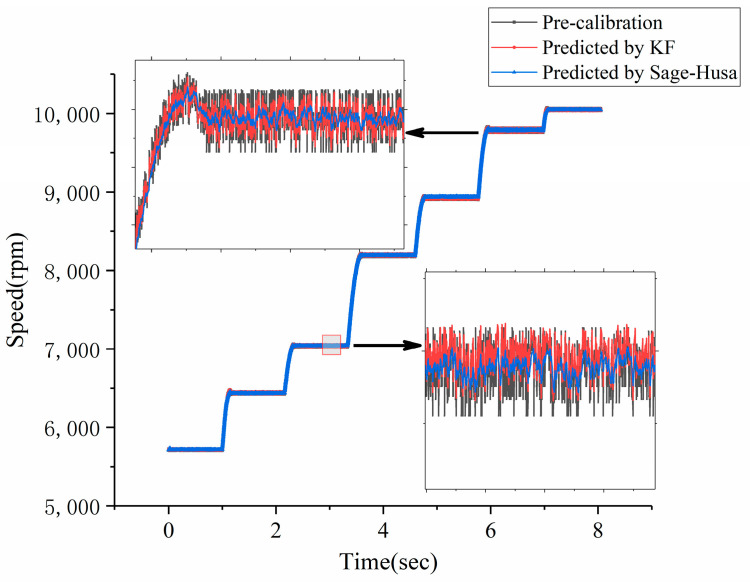
The speed prediction results are compared among the pre-calibration correction, Kalman Filter (KF), and Sage–Husa method for the motor speed range between 5000 rpm and 10,000 rpm.

**Figure 11 sensors-23-06604-f011:**
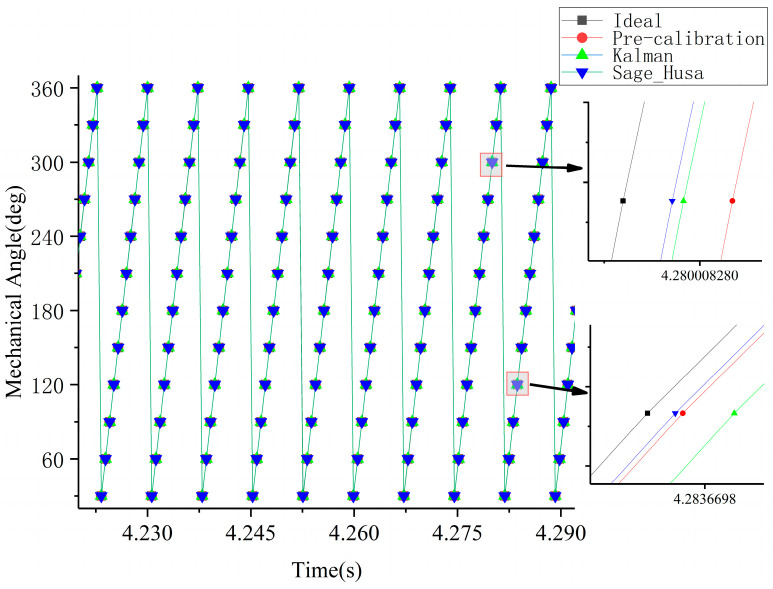
Relationship between measured phase shift moment, ideal phase shift moment, predicted phase shift moment by KF, and predicted phase shift moment by Sage–Husa method.

**Table 1 sensors-23-06604-t001:** Partial parameters of each sensor.

Sensors Parameters	Value
SZMB-10 Measurement Range	0–20,000 Hz
LA25-P Measurement range	0–25 A
LA25-P Error	±0.65%

**Table 2 sensors-23-06604-t002:** Phase difference between the actual edge and the ideal edges.

Speed (rpm)	$HallAF_{1}$ (%)	$HallAR_{1}$ (%)	$HallAF_{2}$ (%)
960	1.67	0.03	0.22
1107	1.54	0.03	0.22
1330	1.46	0.03	0.23
Weighted Sum	1.59	0.03	0.22

**Table 3 sensors-23-06604-t003:** Correction value of relative deviation for each rising edge.

Speed (rpm)	A1 (°)	B1 (°)	C1 (°)	A2 (°)	B2 (°)	C2 (°)
960	−0.906	0.031	0.875	−0.772	−0.182	0.955
1107	−0.935	0.028	0.933	−0.803	−0.193	0.996
1330	−0.980	0.027	0.953	−0.819	−0.165	0.985
Weighted Sum	−0.930	0.021	0.908	−0.791	−0.182	0.974

**Table 4 sensors-23-06604-t004:** Phase difference between the falling stage of the $E_{BC}$ zero-crossing point with HallAR.

Speed (rpm)	$HallAR_{1}$ (%)	$HallAR_{2}$ (%)
960	4.25	4.22
1107	4.21	4.19
1330	4.18	4.17
Weighted Sum	4.22	4.20

**Table 5 sensors-23-06604-t005:** Comparison of prediction results from three different methods.

	The MAPE for Regulation Speed Stage (%)	The Maximum Overshoot of Speed (%)	The MAPE for Constant Speed Stage (%)	The MAPE of the Commutation Time (%)
Pre-calibration	0.7500	0.0456	0.0101	0.0087
KF method	0.6700	0.0348	0.0082	0.0077
Sage–Husa method	0.3700	0.0153	0.0037	0.0066

## Data Availability

Not applicable.

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
