# Peer review of "A Sage–Husa Prediction Algorithm-Based Approach for Correcting the Hall Sensor Position in DC Brushless Motors"

_sensors, 2023, doi:10.3390/s23146604_

Round 1
Reviewer 1 Report
The manuscript have acceptable originality, and overall presentation is fine too. However, few observations need author attention to revise the manuscript
1) The last paragraph of the introductory part must include the organization of the manuscript
2) Author is suggested to add a remark in the introductory part that will summarize the novelty of work.
3) Some Figures need to be more vibrant, such as Fig. 2, 3.
4) Grammatical editing from a native speaker is required.
Already given in the author comments section
Reviewer 2 Report
The authors need to consider the following suggestions:
(1) The introduction needs to be improved by discussing current articles, as only a few new articles have been included in it.
(2) The quality of all figures needs to be increased, and the size of the labels should be enlarged as well.
(3) It is mandatory for the authors to compare their results with other proposals reported in the literature in order to validate or demonstrate the importance of their contribution to the subject.
(4) A new section named "results discussion" needs to be added to condense all the obtained results. I recommend presenting a table that showcases the qualitative and quantitative features of your proposal and the other reviewed proposals, to highlight the main advantages and disadvantages of your approach.
(5) The conclusion needs to be improved by including quantitative results, not just qualitative results. Additionally, it is important to mention the next steps in the investigation.
Reviewer 3 Report
This paper addresses an important industrial issue, the control of BLDCM motors. Nevertheless, I find serious drawbacks in this paper. The most important one is the mismatch between the title and the content of the paper. The title focuses on a "Sage-Husa prediction algorithm-based algorithm", but the paper focuses mainly in calibrating the signals from the Hall sensors, The Sage-Husa method is just briefly commented on page 9, and on page 16 it is compared with the Kalman filtering approach just saying that is that it reduces the speed fluctuations by 49 % (line 595). No data is presented to support this assertion. Furthermore, in Fig. 11 very small variations appear when comparing both methods. Therefore, the conclusions are not supported by the presented data.
In the experimental section the majority of the results are related with the calibration of the motor used in this paper. The proposed techniques are well known in the technical literature, and cannot considered an innovation of this paper.
Form a methodological point of view, the presentation is confusing, with many typos in the text, hard to read figures and hard to follow formulation, which could be substituted with appropriate references to existing literature. Also, many references are bad formatted (Ieee instead of IEEE) or are not available to the general public, being internal publications of universities.
I do not see a clear innovation in this paper, which, along with the deficiencies in the presentation and the methodology, make me reject this paper in its present form.
This paper has serious flaws in the presentation, methodology and results that must be addressed before being considered for publication.
Reviewer 4 Report
This manuscript covers an important subject of the rotor position is essential for the control of Brushless DC motors.
The manuscript is very well written, the methodology is well presented, as well as the practical implementation and obtained results with the analysis. The conclusions made are very well supported by the results presented. The list of references is appropriate, as it contains most of the important references from the matter discussed.
I have only some technical remarks that need to be considered, in particular:
1. The symbols, numbers and text in most of the figures need to be larger. The quality of figures needs to be better to eliminate blurriness.
2. Line 2790, probably N1 should be written instead of PN1 at the beginning, which would be in accordance with Figure 2. Check this.
3. The term "comprehensive" in the caption of Figure 4 might be omitted or changed, as it sounds subjective.
4. It is not clear what is the point of section 2.4. Please, revise it or delete if not needed.
5. Line 320, this subtitle should be on the next page, along with the corresponding text.
6. Tables 3 and 4 need to be discussed in more details.
7. It sounds like that the part 4. and the second paragraph of the part 5. on page 14 should be extended by an additional discussion.
8. Places of Figures 8 and 9 within the corresponding text are mixed.
9. Line 546, the term "ideal" is not appropriate, maybe "symmetrical" or something similar would be better.
10. Line 548, particular numerical results of the power consumption would be of very significant importance in this place. Consider this.
11. I do not find the Supplement so important, but it can stay. The number of figures is not correct.
Round 2
Reviewer 2 Report
The authors have correctly addressed the reviewer's concerns. Hence, I can recommend accepting the article.
Reviewer 3 Report
The Authors have addressed adequately my previous concerns. Moderate editing of English language required. I find this paper suitable for publication in its present form.
Some typos must be corrected before publication.